



# EVAPORATION FRONT AND ITS MOTION

Jiří Mls[1]

[1]Charles University, Faculty of Science, Albertov 6, 128 43 Praha 2, Czech Republic ,

**Correspondence:** Jiří Mls (jiri.mls@natur.cuni.cz)

**Abstract.** The evaporation demands upon a rock or soil surface can exceed the ability of the profile to bring sufficient amount of liquid water. A dry surface layer arises in the porous medium that enables just water vapor flow to the surface. The interface between the dry and wet parts of the profile is known as the evaporation front.

The paper gives the exact definition of the evaporation front and studies its motion. A set of differential equations governing
the front motion in space is formulated. Making use of a set of measured and chosen values, a problem is formulated that illustrates the obtained theory. The problem is solved numerically and the results are presented and discussed.

**Keywords.** Evaporation front, evaporation front motion, characteristic surface layer, rate of evaporation

## 1 Introduction

Under arid or semiarid conditions, the evaporation demands usually exceed the ability of the exposed porous medium to
provide liquid-phase water. The water content of subsurface zones of coarse-grained rocks or sandy soils is usually very below its residual value and consequently only the gas-phase water can flow through. These, mostly up to few centimeters thick, zones are referred to as vapor zones, dry surface layers or evaporation zones (e.g. Yamanaka and Yonetani, 1999; Saravanapavan and Salvucci, 2000; Shokri et al., 2009; Deol et al., 2014).

The extent and development of the dry surface layer significantly affect the material's decay: directly by changes in its
wetness, by frost and particularly by salt weathering, since dissolved salts are transported by the capillary water and form crystals at places of evaporation (e.g. Rijniers et al., 2005; Mol and Viles, 2013; Kurtzman et al., 2016).

The phenomenon is preconditioned by the fact that the porous medium becomes impervious for the liquid-phase water if the water content becomes sufficiently small. A limit arises inside the porous medium behind which the transport of water is only possible in the form of vapor. Such a soil profile can be divided into two parts that can be referred to as the zone of water
flow and the zone of vapor diffusion. This formulation, however, is too vague and leaves the intermediate zone, its extend and nature, unclear.

Several studies were published giving detailed description of the evaporation process and the development of the transition zone (e.g. Lehmann et al., 2008; Shokri et al., 2010; Sakai et al., 2011; Or et al., 2013; Assouline et al., 2013; Rothfuss et al., 2015). These papers are mostly focused to special problems, unlike the present paper which studies the 3D problem under
general transport conditions.





Sakai et al. (2011) studied the problem considering both hydraulic and thermal processes and detected a narrow transition layer at the bottom of the dry surface layer. Another approach, (Konukcu et al., 2004), connects the interface between the region of water flow and the region of vapor flow, denoted as the evaporation front, with a critical value of the water content that can be determined directly from such porous medium characteristics as hydraulic conductivity and vapor diffusivity.

Hadley (1982) studied the problem of water vapor transport through a region of dry material from a receding evaporation front. In the paper, the heat balance equation was involved to the final system of equations, the evaporation front was considered as a sharp interface between the saturated zone and the dry (without liquid water) zone; the front was fixed and given a-priori, and the liquid water was unmovable.

Kulikovskii (2002) studied the time-space development of discontinuities in one-dimensional porous medium. Liquid water,
vapor and mixture of liquid water and vapor were assumed in the void space and two governing equations, water and heat flow, were considered. No particular interfaces were defined, discontinuities, in general, were studied in the time-space. Il'ichev and Shargatov (2013) started their study with similar assumptions concerning the governing laws and investigated the resulting transition surfaces and conditions of loss of their stability.

Unlike these studies, the present paper aims to define the evaporation front by means of porous media characteristics and to
formulate the law of its motion generally not involving any particular law governing the water transport. This approach makes it possible to use any set of flow and transport laws when formulating a problem of the evaporation front motion.

Several methods using dyes were developed to visualize the dry and wet regions within soil or rock profiles (e.g. Shokri et al., 2009; Bruthans et al., 2018; Kumar and Arakeri, 2018; Weiss et al., 2018). These methods proved their efficiency in laboratory conditions when utilized to visualize the dry surface layers and, in particular, the evaporation front positions. The
applied water-dye solutions increase their concentration at places of evaporation and indicate these places by changing their color.

A special method was developed (Weiss et al., 2020) that minimizes the medium destruction and is usable under the field conditions. A very thin rod covered by a layer of color is inserted into a narrow hole drilled to the investigated material where there is the sought evaporation front. The present liquid-phase water colors the corresponding part of the rod showing its
extend.

Numbers of experiments aimed at seeking for and visualizing the evaporation front, see (Weiss et al., 2018) and (Weiss et al., 2020), show that its position can be detected as a sharp line. The present paper tries to respect this experimental result in the definition presented below.

The goal of this paper is to give an exact definition of the evaporation front and to formulate the law of its motion.

**2    Basic assumptions and theory**

We study such processes of water transport in porous media, where both the fluid phases, gaseous and liquid, are present and evaporation is taken into account. A porous medium domain is considered that is in contact with a wet neighborhood at a part of its boundary while at the other part of the boundary, it is in contact with a dry neighborhood. Here wet and dry is understand





as containing liquid water and without liquid water, respectively. Such a part of the domain's boundary which is open to the
atmosphere is considered as the dry contact.

Under these conditions, there necessarily exists a set of points inside the studied domain or upon its boundary that makes
an interface between the wet medium (porous medium) and the dry medium (porous medium or air). In view of the above
introduced terms, these points can be considered as points of the evaporation front.

Generally, the porous medium profile can be divided into three parts: (a) the dry zone, where just two phases, solid and
gaseous (air), are present and water exists in the form of vapor as a component of the gaseous phase, (b) the wet zone, where
the movable liquid water exists, and (c) the intermediate zone, where the liquid water is present but only in such a contact
with the solid phase, that makes it unmovable. Here, such liquid phase water is understood as movable that moves due to the
hydraulic head gradient.

The evaporation front does not exist in itself; it is a matter of definition. It seems natural to place the evaporation front to
the intermediate zone or to an interface between the intermediate zone and one of the neighboring zones. It can be expected
that during the process of evaporation, the depth of the intermediate zone becomes small. The present water evaporates quickly
due to its immobility, its small amount and contact with the solid phase. In view of this and of the fact that experimentally the
evaporation front can be indicated as a sharp interface between two neighboring zones, we assume: the extend of the zone (c)
can be neglected and the evaporation front is defined as the common boundary of the zone without liquid water and the zone
with movable liquid water. The concept evidently enables existence of a jump in water content values.

We do not consider the temperature distribution and heat flow and balance; it is only supposed that the heat conditions within
the profile are sufficient to provide the amount of the latent heat of vaporization that is necessary for the evaporation resulting
from the actual process of water transport.

In virtue of its definition, the evaporation front changes its position with time according to the outer conditions. Its shape
and motion results from mutual relations (water transfer) between the wet zone and the dry zone. The front moves towards the
wet region if the evaporation exceeds the flow of the liquid water towards the interface through the wet zone and vice versa.
Since the evaporation front inside porous media, e.g. in a rock massif, is difficult to detect, mathematical modelling becomes
an important tool always if the knowledge of its position and motion is required.

In what follows, all the introduced characteristics are macroscale porous-medium characteristics; e.g. a domain is a macroscale
domain, a surface is a macroscale surface, etc..

Denote by $\Omega$, $\Omega \subset \mathbb{R}^3$, the domain in space, and by $(0, T)$ the time interval in which we study the transport process and
suppose that the movable liquid-phase water occupies an open part $G^w$ of the time-space domain $G$ (i.e. the water content $\theta$ is
positive and sufficient to enable the water flow in $G^w$), where

$$G = (0, T) \times \Omega.$$

We further define

$$G^d = G \smallsetminus \overline{G^w};$$

in virtue of our assumptions, $\theta = 0$ in $G^d$.





To any time $t \in (0,T)$ we define the wet zone $\Omega_t^w$ and the dry zone $\Omega_t^d$ by putting

$$\Omega_t^w = \{\boldsymbol{x} \in \mathbb{R}^3 \,;\, (t,\boldsymbol{x}) \in G^w\} \quad \text{and} \quad \Omega_t^d = \{\boldsymbol{x} \in \mathbb{R}^3 \,;\, (t,\boldsymbol{x}) \in G^d\}. \tag{1}$$

It holds

$$\Omega_t^w \cap \Omega_t^d = \emptyset, \quad \overline{\Omega_t^w} \cup \overline{\Omega_t^d} = \overline{\Omega} \quad \text{for } t \in (0,T). \tag{2}$$

We define the evaporation front $\gamma_t$ at time $t \in (0,T)$ as

$$\gamma_t = \overline{\Omega_t^w} \cap \overline{\Omega_t^d}$$

This definition covers the wet-dry interfaces inside $\Omega$. In order to enable the location of the evaporation front upon the
domain's surface, we define the wet and dry contacts from outside of $\Omega$. The sets $\Omega_t^w$ and $\Omega_t^d$ divide the boundary $\partial\Omega$ into wet
and dry parts with respect to wet-dry conditions inside $\Omega$ at $t \in (0,T)$. Denote by $B_t^w$ and $B_t^d$ such two parts of $\partial\Omega$ that are
at time $t$ in contact with wet and dry conditions outside $\overline{\Omega}$, respectively, and define: a boundary point $\boldsymbol{x} \in \partial\Omega$ belongs to the
evaporation front at time $t$ if it satisfies

$$\boldsymbol{x} \in \overline{\Omega_t^w} \cap B_t^d \quad \text{or} \quad \boldsymbol{x} \in \overline{\Omega_t^d} \cap B_t^w.$$

The complete evaporation front $\gamma_t$ at time $t \in (0,T)$ is then

$$\gamma_t = \left(\overline{\Omega_t^w} \cap \overline{\Omega_t^d}\right) \cup \left(\overline{\Omega_t^w} \cap B_t^d\right) \cup \left(\overline{\Omega_t^d} \cap B_t^w\right).$$

The image of the evaporation front in time-space is

$$\Gamma = \{(t,\boldsymbol{x}) \in \mathbb{R}^4 \,;\, \boldsymbol{x} \in \gamma_t, \; t \in (0,T)\}.$$

We assume that $\Gamma$ is a smooth hypersurface in $\mathbb{R}^4$ and denote by $\boldsymbol{\nu}^\Gamma \in \mathbb{R}^4$ the unit vector normal to $\Gamma$ that points out of the wet
part or into the dry part of $G$. Since we defined the wet and dry boundary $B_t^w$ and $B_t^d$ of $\Omega$ at time $t$ with respect to the outer
conditions, this orientation has sense everywhere on $\Gamma$.

Suppose that $(t,\boldsymbol{\xi}) \in \Gamma$ and

$$\nu_t^\Gamma(t,\boldsymbol{\xi}) \neq 0,$$

where $\boldsymbol{\nu}^\Gamma = (\nu_t^\Gamma, \nu_1^\Gamma, \nu_2^\Gamma, \nu_3^\Gamma)$. Then the hypersurface $\Gamma$ can be in a certain neighborhood of $(t,\boldsymbol{\xi})$ expressed by a function $\tau$ in
the form of equation

$$t = \tau(x_1, x_2, x_3).$$

Then $\nu_t^\Gamma < 0$ implies the existence of a positive value $\tau$ such that $(t+\vartheta, \boldsymbol{\xi}) \in G^w$ for $\vartheta \in (0,\tau)$. Consequently, the evaporation
front $\gamma_t$ moves at its point $\boldsymbol{\xi}$ towards the dry zone if $\nu_t^\Gamma$ is negative at $(t,\boldsymbol{\xi})$ and vice versa.





The position of the evaporation front results from the mutual relations between the water transport in the wet zone and in the dry zone. Denote by $\boldsymbol{w}$ the volumetric flux density of liquid water in the wet zone, by $\boldsymbol{v}^d$ and $\boldsymbol{v}^w$ the volumetric flux density of the gaseous phase in the dry zone and in the wet zone, and by $\boldsymbol{b}^d$ and $\boldsymbol{b}^w$ the water vapor flux density by diffusion in the gaseous phase within the dry zone and the wet zone. Let further $c^d$ and $c^w$ denote the water vapor concentration in the gaseous phase within the dry zone and the wet zone. We suppose that functions $\boldsymbol{b}^d, c^d, \boldsymbol{v}^d$ are continuous in $\overline{G^d}$ and functions $\boldsymbol{b}^w, c^w, \boldsymbol{v}^w, \boldsymbol{w}$ are continuous in $\overline{G^w}$.

The evaporation front is not connected with a fixed set of mass points and the problem of its motion is not a problem of the particle tracking. The evaporation front moves in such a direction and with such a velocity that are given by the balance of mass of water. Since the tangential motion of the evaporation front at its point does not change the front's position, the evaporation front moves at each point in the direction of its normal.

Let $\boldsymbol{\xi} = \boldsymbol{\xi}(t)$ be a point upon the evaporation front at time $t$; $\boldsymbol{\xi} \in \gamma_t$. Let $\delta t$ be an elementary time step and $\delta S$ an elementary surface surrounding the point $\boldsymbol{\xi}$, $\delta S \subset \gamma_t$. Denote by $\boldsymbol{\nu}(t, \boldsymbol{\xi})$ the unit normal vector to $\gamma_t$ at point $\boldsymbol{\xi}$, oriented out of the wet zone, and by $\delta s$ the distance between the positions $\gamma_t$ and $\gamma_{t+\delta t}$ at $\boldsymbol{\xi}$. Then the next position $\boldsymbol{\xi} + \delta s \boldsymbol{\nu}(t, \boldsymbol{\xi}) \in \gamma_{t+\delta t}$ of point $\boldsymbol{\xi}$ at time $t + \delta t$ satisfies the balance equation

$$
\delta S \, \delta s \, n \, c^d(t, \boldsymbol{\xi}) + \delta S \left( \left( \rho \boldsymbol{w}(t, \boldsymbol{\xi}) + c^w(t, \boldsymbol{\xi}) \boldsymbol{v}^w(t, \boldsymbol{\xi}) + \boldsymbol{b}^w(t, \boldsymbol{\xi}) \right), \boldsymbol{\nu}(t, \boldsymbol{\xi}) \right) \delta t -
$$
$$
\delta S \left( \boldsymbol{b}^d(t, \boldsymbol{\xi}) + c^d(t, \boldsymbol{\xi}) \boldsymbol{v}^d(t, \boldsymbol{\xi}), \boldsymbol{\nu}(t, \boldsymbol{\xi}) \right) \delta t = \delta S \, \delta s \left( c^w(t, \boldsymbol{\xi})(n - \theta(t, \boldsymbol{\xi})) + \rho \theta(t, \boldsymbol{\xi}) \right), \tag{3}
$$

where values of higher orders of $\delta t$ and $\delta s$ are neglected. The idea of this calculation can be seen in Figure 1. It shows two positions of the front, at times $t$ and $t + \delta t$, and two possible vectors, $\boldsymbol{W}$ and $\boldsymbol{B}$, where, for the sake of simplicity, $\boldsymbol{B}$ denotes the vector sum of fluxes by diffusion

$$
\boldsymbol{B} = \boldsymbol{b}^d(t, \boldsymbol{\xi}) - \boldsymbol{b}^w(t, \boldsymbol{\xi}),
$$

and $\boldsymbol{W}$ denotes fluxes due to water flow and advection by the gaseous phase motion

$$
\boldsymbol{W} = \rho \boldsymbol{w}(t, \boldsymbol{\xi}) + c^w(t, \boldsymbol{\xi}) \boldsymbol{v}^w(t, \boldsymbol{\xi}) - c^d(t, \boldsymbol{\xi}) \boldsymbol{v}^d(t, \boldsymbol{\xi}).
$$

The depicted directions of these vectors suggest simultaneous flooding of the dry zone and drying of the wet zone. These two processes act against each other; the figure shows that the vapor transport predominates and the front moves towards the wet zone.

The limit form of Eq. (3) for $\delta t$ approaching to zero is

$$
\frac{\mathrm{d}s}{\mathrm{d}t}(t, \boldsymbol{\xi}) = \frac{\left( \left( \rho \boldsymbol{w}(t, \boldsymbol{\xi}) + c^w(t, \boldsymbol{\xi}) \boldsymbol{v}^w(t, \boldsymbol{\xi}) + \boldsymbol{b}^w(t, \boldsymbol{\xi}) - \boldsymbol{b}^d(t, \boldsymbol{\xi}) - c^d(t, \boldsymbol{\xi}) \boldsymbol{v}^d(t, \boldsymbol{\xi}) \right), \boldsymbol{\nu}(t, \boldsymbol{\xi}) \right)}{c^w(t, \boldsymbol{\xi})(n - \theta(t, \boldsymbol{\xi})) - n \, c^d(t, \boldsymbol{\xi}) + \rho \theta(t, \boldsymbol{\xi})} \tag{4}
$$

and hence

$$
\frac{\mathrm{d}\boldsymbol{\xi}}{\mathrm{d}t}(t) = \frac{\left( \left( \rho \boldsymbol{w}(t, \boldsymbol{\xi}) + c^w(t, \boldsymbol{\xi}) \boldsymbol{v}^w(t, \boldsymbol{\xi}) + \boldsymbol{b}^w(t, \boldsymbol{\xi}) - \boldsymbol{b}^d(t, \boldsymbol{\xi}) - c^d(t, \boldsymbol{\xi}) \boldsymbol{v}^d(t, \boldsymbol{\xi}) \right), \boldsymbol{\nu}(t, \boldsymbol{\xi}) \right)}{c^w(t, \boldsymbol{\xi})(n - \theta(t, \boldsymbol{\xi})) - n \, c^d(t, \boldsymbol{\xi}) + \rho \theta(t, \boldsymbol{\xi})} \boldsymbol{\nu}(t, \boldsymbol{\xi}). \tag{5}
$$





Equation (5) is the governing equation of the evaporation front motion in the interval $(0, T)$. Note that, in view of the introduced assumptions, the values of functions $\theta, \boldsymbol{w}, \boldsymbol{b}^d, \boldsymbol{b}^w, c^d, c^w, \boldsymbol{v}^d$ and $\boldsymbol{v}^w$ on $\Gamma$ are uniquely defined. The water density and porosity are presented as constants in Eqs (3) and (5). Such an assumption is evidently not necessary and the equations remain unchanged

for $n = n(t, \xi)$ and $\rho = \rho(t, \xi)$.

In the cases we commonly meet in connection with problems of evaporation, the flow of the gaseous phase is restricted to balancing the changing volume of liquid water, i.e.

$$\|\boldsymbol{w}\| \approx \|\boldsymbol{v}^d\| \approx \|\boldsymbol{v}^w\|$$

and since

$\rho \|\boldsymbol{w}\| \gg c^d \|\boldsymbol{v}^d\| \quad \text{and} \quad \rho \|\boldsymbol{w}\| \gg c^w \|\boldsymbol{v}^w\|,$

the advective transport of water vapor can be neglected. The governing equation becomes

$$\frac{\mathrm{d}\boldsymbol{\xi}}{\mathrm{d}t}(t) = \frac{\left( (\rho \boldsymbol{w}(t, \boldsymbol{\xi}) + \boldsymbol{b}^w(t, \boldsymbol{\xi}) - \boldsymbol{b}^d(t, \boldsymbol{\xi})), \boldsymbol{\nu}(t, \boldsymbol{\xi}) \right)}{c^w(t, \boldsymbol{\xi})(n - \theta(t, \boldsymbol{\xi})) - n\, c^d(t, \boldsymbol{\xi}) + \rho\, \theta(t, \boldsymbol{\xi})} \, \boldsymbol{\nu}(t, \boldsymbol{\xi}). \tag{6}$$

## 3  Problem formulation

The front's motion reflects the proportions between the water flow and transport out of the front and towards the front which

are given by the laws of flow and transport in the wet zone and in the dry zone. In order to evaluate flow and transport in porous media, Darcy's law and Fick's law can be utilized

$$\boldsymbol{w} = -k(h)\,\mathrm{grad}\,(x_3 + h), \quad \boldsymbol{b}^d = -D^d\,\mathrm{grad}\,c^d, \quad \text{and} \quad \boldsymbol{b}^w = -D^w(\theta)\,\mathrm{grad}\,c^w, \tag{7}$$

where the third coordinate $x_3$ is oriented vertically upwards, $h$ is the pressure head, $k$ is the hydraulic conductivity, and $D^d$ and $D^w$ are the coefficients of water diffusion in air within the porous medium.

In the dry zone, water is present in the form of water vapor and its motion is governed by the continuity equation with the use of Fick's law:

$$\frac{\partial}{\partial t}(n\,c^d) - \frac{\partial}{\partial x_i}\left( D^d \frac{\partial c^d}{\partial x_i} \right) + \frac{\partial}{\partial x_i}(c^d\, v_i^d) = 0. \tag{8}$$

The vapor motion in the wet zone is governed by the same laws

$$\frac{\partial}{\partial t}\left( (n - \theta)\, c^w \right) - \frac{\partial}{\partial x_i}\left( D^w(\theta) \frac{\partial c^w}{\partial x_i} \right) + \frac{\partial}{\partial x_i}(c^w\, v_i^w) = 0, \tag{9}$$

and the motion of liquid phase water in the wet zone is governed by Richards' equation

$$\frac{\partial \theta}{\partial t} - \frac{\partial}{\partial x_i}\left( k(h)\left( \frac{\partial x_3}{\partial x_i} + \frac{\partial h}{\partial x_i} \right) \right) = 0. \tag{10}$$





In virtue of the introduced theory, the evaporation front motion is governed by Eqs. (8), (9), (10) and (5) or (6). The unknown functions are $\theta$ and $h$, connected by the retention curve, $c^w$, $c^d$ and $\xi$ defined in $\overline{G^w}$, $\overline{G^d}$ and $[0, T]$. The functions $n, k, D, \boldsymbol{v}$ and $\rho$ are supposed to be known or given by additional equations.

In this way, Eqs. (9) and (10) in $G^w$ and Eq. (8) in $G^d$ stand for a coupled moving boundary problem, and Eq. (5) is a condition imposed upon the movable common part of the boundaries of the domains. The unknown function $\xi$ defined in $[0, T)$ is then given as the position of the moving boundary.

Another possible formulation of the problem is to solve the ordinary differential equation (5) in the interval $[0, T]$, where the right-hand side of the equation is given as the solution of the problems (9), (10) and (8) defined in $G^w$ and $G^d$.

The latter approach was utilized when solving the problem presented in the 5-th section.

## 4   One-dimensional problem

Let now the studied domain be an interval $\Omega \subset \mathbb{R}^1$,

$$\Omega = (0, L), \quad \Omega_t^w = (0, \xi(t)), \quad \Omega_t^d = (\xi(t), L),$$

where the function $\xi(t)$ that denotes the position of the evaporation front at time $t$ is a scalar function defined in $(0, T)$. The sets $G^w$ and $G^d$ are

$$G^w = \{(t, x) \in \mathbb{R}^2 \,; t \in (0, T), \ x \in (0, \xi(t))\} \quad \text{and} \quad G^d = \{(t, x) \in \mathbb{R}^2 \,; t \in (0, T), \ x \in (\xi(t), L)\}.$$

The image of the evaporation front in time-space is

$$\Gamma = \{(t, x) \in \mathbb{R}^2 \,; \xi(t) - x = 0, \ t \in (0, T)\}$$

and $\boldsymbol{\nu}^\Gamma \in \mathbb{R}^2$, the unit normal to $\Gamma$, is

$$\boldsymbol{\nu}^\Gamma(t, \xi(t)) = \left( -\frac{\mathrm{d}\xi}{\mathrm{d}t}(t), 1 \right) \Big/ \sqrt{\left(\frac{\mathrm{d}\xi}{\mathrm{d}t}(t)\right)^2 + 1}.$$

The above presented assertion concerning the direction of the evaporation front motion with respect to the $\mathrm{sgn}\,(\nu_t^\Gamma)$ is now evident.

In virtue of the introduced theory, the law of the evaporation front motion is given by Eq. (5). In one space dimension, since $\nu(t, \xi)$ is either 1 or -1, the equation reads

$$\frac{\mathrm{d}\xi}{\mathrm{d}t}(t) = \frac{\rho\, w(t, \xi) + c^w(t, \xi)\, v^w(t, \xi) + b^w(t, \xi) - b^d(t, \xi) - c^d(t, \xi)\, v^d(t, \xi)}{c^w(t, \xi)(n - \theta(t, \xi)) - n\, c^d(t, \xi) + \rho\, \theta(t, \xi)}$$

and its simplified form (6) is

$$\frac{\mathrm{d}\xi}{\mathrm{d}t}(t) = \frac{\rho\, w(t, \xi) + b^w(t, \xi) - b^d(t, \xi)}{c^w(t, \xi)(n - \theta(t, \xi)) - n\, c^d(t, \xi) + \rho\, \theta(t, \xi)}. \tag{11}$$

In order to complete the problem formulation, to determine the right-hand side function, the one-dimensional form of equations (6) to (10) can be utilized.





## 5   A solved problem

In the frame of a wider research that concerns evaporation from rock surfaces and the time dependence of evaporation front positions, an experiment was carried out. It was an unpublished auxiliary experiment performed by several of the author's colleagues and since it was not sufficiently documented from the point of view of this study, it cannot be simulated. Nevertheless, part of its results can be utilized here in order to present an example of possible use of the achieved theoretical results. The missing data were simply chosen, not optimized. The following description should be understood as a problem formulation not as a documentation of measurements.

A cylinder shaped sample of the studied rock was put to the position with horizontal axis. The jacket of the cylinder was insulated so that no water (of any phase) could penetrate and the motion of water through the sample was possible only in the horizontal direction along the cylinder's axis.

The length of the sample was $L = 49$ mm, and the length of the time interval was $T = 63$ days. One open end of the sample, say $x = 0$, was equipped so that it was possible to measure the pressure head and the rate of water inflow into the sample at this point. The obtained discrete data were approximated by smooth functions $h_0(t)$ and $w_0(t)$ (pressure head and volumetric flux density), $h_0, w_0 \in C^1(0, T)$, that were utilized as the imposed boundary conditions.

Figure 2 shows the pressure head values; the squares are the measured data and the solid curve is their approximation $h_0$.

The volumetric flux density was measured indirectly in the form of discrete values of the cumulative flux to the sample. In Figure 3, the step function represents the measured data and the smooth function is the boundary condition $w_0$. The integral values over $(0, T)$, (the total inflow to the sample per unit surface) of both functions are equal. The other end of the sample, $x = L$, was left open to the outer (atmospheric) conditions which were not measured.

Soil moisture retention data and the hydraulic conductivity (at saturation) $K_s$ were obtained elsewhere using samples similar material. Making use of these characteristics, the Mualem and van Genuchten parameters $\theta_r, \theta_s, \alpha, m$ and $n$ were determined and hence, the hydraulic conductivity and the soil moisture retention curve as functions of either pressure head or water content were defined.

Similarly the value of the diffusion coefficient of water in the gaseous phase within the porous medium, $D^d$, was obtained measured on samples of the utilized material. The characteristic surface layer $\varepsilon$, was added from outside to the domain $\Omega$, its value was taken from the paper (Slavík et al., 2020), where the layer is referred to as "calibrated $L$" and also as "boundary layer".

The fluorescein visualization method, (Weiss et al., 2018), was utilized to detect the front's position during the experiment. Fluorescein was applied at the water input side of the sample and a set of couples $(t, \xi)$, time and the front's position, was registered. The data were calibrated (using a simple linear transformation) to agree with the value measured after finishing the experiment.

The following problem was formulated. In the interval $[0, T]$, the solution $t \mapsto \xi(t), \xi \in C^1(0, T) \cap C[0, T]$, of equation (11) is sought that satisfies the initial condition

$$\xi(0) = L, \tag{12}$$





where $\rho$ and $n$ (density and porosity) are known constants.

Since neither functions $c^d$ and $c^w$ nor their boundary values at $x = 0$, $\xi(t)$ and $L$ were measured during the experiment, the following two assumptions were introduced.

(*) The vapor concentration has a constant value $c_f$ at the evaporation front in the gaseous phase and at points of positive value of water content $\theta$, i.e. at points of contact with liquid phase water.

(**) There are steady-state outer atmospheric conditions during the process giving a constant value $c_e$ of vapor concentration

in the gaseous phase at $x = L$.

The assumptions do not contradict each other at the point $\xi(0)$, since the characteristic surface layer $\varepsilon$ was accepted in the model. As presented above, the right-hand side of Eq. (11) can be determined by solving the related initial-boundary value problems with Eqs. (8), (9), (10).

In view of assumption (*) and Eqs. (9) and (7), it holds

$$c^w(t,x) = c_f \quad \text{and} \quad b^w(t,x) = 0 \qquad \text{in } G^w.$$

Applying assumption (*), Eq. (9) reads

$$\frac{\partial \theta}{\partial t} - \frac{\partial v^w}{\partial x} = 0.$$

The comparison with Richards' equation (10) gives $v^w = -w$; the gaseous phase continuously replaces the leaving water. Similar result has already been expected even for more general cases, see the estimations utilized when replacing Eq. (5) by

Eq. (6). Making use of these results, Eq. (11) becomes

$$\frac{d\xi}{dt}(t) = \frac{\rho w(t,\xi) - b(t,\xi)}{\theta(t,\xi))(\rho - c_f)} \tag{13}$$

where the unknown parameters at the right-hand side of (13) are solutions of the following two initial-boundary value problems and, since $b^w$, $c^w$ and $D^w$ do not appear in what follows, $b$, $c$ and $D$ stand for $b^d$, $c^d$ and $D^d$.

To determine functions $\theta$ and $w$, function $h$ defined in $G^w$ is sought that satisfies equation (10), now in the form

$$\frac{\partial \theta}{\partial t} - \frac{\partial}{\partial x}\left(k(h)\frac{\partial h}{\partial x}\right) = 0, \tag{14}$$

and the conditions

$$h(0,x) = h_i(x) \qquad \text{in} \quad (0,L),$$

and

$$h(t,0) = h_0(t) \quad \text{and} \quad -k(h(t,0))\frac{\partial h}{\partial x}(t,0) = w_0(t) \quad \text{in } (0,T), \tag{15}$$

where $\theta(h)$, is the retention curve. Functions $h_0$ and $w_0$ are known from the experiment and function $h_i$, the initial pressure head distribution, was not measured and has to be determined.





Incorporating the characteristic surface layer to the problem formulation, we change the set $G^d$ to

$$G^d = \{(t,x) \in \mathbb{R}^2 \,; x \in (\xi(t), L+\varepsilon),\, t \in (0,T)\}.$$

Now, $c$ solves the equation

$$\frac{\partial}{\partial t}(n c) - \frac{\partial}{\partial x}\left(D \frac{\partial c}{\partial x}\right) = 0 \tag{16}$$

in $G^d$ and satisfies the conditions

$$c(0,x) = c_i(x) \quad \text{in} \quad (L, L+\varepsilon),$$

and

$$c(t, \xi(t)) = c_f \quad \text{and} \quad c(t, L+\varepsilon) = c_e \quad \text{in} \ (0,T), \tag{17}$$

where $c_i$ is the initial distribution of the water concentration in the gaseous phase. The values $c_e$ and $c_f$ were chosen with respect to these requirements: the relative humidity at the front was 100%, the outer relative humidity was between 30% and 50% and the temperature between 18°C and 23°C.

In order to define the initial state of the sample, functions $c_i$ and $h_i$, it was assumed: the process started from the steady state water vapor diffusion determined in $(L, L+\varepsilon)$ by the boundary values $c_f$ and $c_e$ and from the steady state water flow determined in $(0,L)$ by the initial conditions $h_0(0)$ and $w_0(0)$. The error involved to functions $c_i$ and $h_i$ vanishes soon, since the sample is small and the imposed boundary conditions take over the dominant role.

Our requirement on the initial conditions is easy to satisfy in the case of function $c_i$ in the domain $\Omega^d_{t=0}$, since, in view of Eq. (16), the governing equation is

$$\frac{\mathrm{d}}{\mathrm{d}x}\left(D \frac{\mathrm{d}c_i}{\mathrm{d}x}\right) = 0 \quad \text{in } (L, L+\varepsilon).$$

With respect to the boundary conditions, the solution reads

$$c_i(x) = \frac{c_e - c_f}{\varepsilon}(x - L) + c_f.$$

In the case of function $h_i$ and domain $\Omega^w_{t=0}$, the governing equation is

$$\frac{\mathrm{d}}{\mathrm{d}x}\left(\rho\, k(h_i) \frac{\mathrm{d}h_i}{\mathrm{d}x}\right) = 0 \quad \text{in } (0,L) \tag{18}$$

with the initial conditions

$$h_i(0) = h_0(0) \quad \text{and} \quad -k(h_i(0)) \frac{\mathrm{d}h_i}{\mathrm{d}x}(0) = w_0(0). \tag{19}$$

The solution to the problem (18), (19) is equivalent to the solution to the problem

$$\frac{\mathrm{d}h_i}{\mathrm{d}x} = -\frac{w_0(0)}{k(h_i)} \quad \text{in } (0,L)$$





with the initial condition $h_i(0) = h_0(0)$. Let the maximum solution to this initial-value problem be defined in $[0, \lambda)$. It can be shown that, in our case, $\lambda > L$ and the wet zone reaches the point $x = L$. Hence, our choice of the initial condition $h_i$ does not contradict the initial condition (12).

Now the problem (13), (12) can be solved numerically. Figure 4 shows three solutions obtained for three different couples of chosen parameters, the boundary values of water concentration in the gaseous phase. The chosen values were

(1) $c_f = 2.27 \times 10^{-5}$ g/cm$^3$, $c_e = 9.35 \times 10^{-6}$ g/cm$^3$,

(2) $c_f = 2.41 \times 10^{-5}$ g/cm$^3$, $c_e = 9.94 \times 10^{-6}$ g/cm$^3$,

(3) $c_f = 2.56 \times 10^{-5}$ g/cm$^3$, $c_e = 1.06 \times 10^{-5}$ g/cm$^3$.

The problem (13), (12) was solved numerically using a predictor-corrector method. The values of the right-hand side were determined using the method of Rothe to solve problems (14) to (15) and (16) to (17).

Let $M_w$ denote the total mass of liquid water in the investigated domain related to unit cross-section. Then it holds

$$M_w(t) = \int_0^{\xi(t)} \rho\,\theta(t,x)\,\mathrm{d}x.$$

Hence, the rate of its change is

$$\frac{\mathrm{d}M_w}{\mathrm{d}t}(t) = \rho\,\frac{\mathrm{d}\xi}{\mathrm{d}t}(t)\,\theta(t,\xi(t)) + \rho\int_0^{\xi(t)} \frac{\partial\theta}{\partial t}(t,x)\,\mathrm{d}x.$$

Making use of Eqs. (13), (14) and (7), we get

$$\frac{\mathrm{d}M_w}{\mathrm{d}t}(t) = \rho\,\frac{\rho\,w(t,\xi(t)) - b(t,\xi(t))}{\rho - c_f} + \rho\big(w(t,0) - w(t,\xi(t))\big).$$

Since $\rho\,w(t,0)$ is the inflow of liquid water into the domain,

$$E(t) = \frac{b(t,\xi(t)) - c_f\,w(t,\xi(t))}{1 - c_f/\rho} \tag{20}$$

is, under the assumptions of this section, the rate of evaporation expressed as the mass of evaporated water per unit time and unit surface of the evaporation front. Consequently, having solved an evaporation-front-motion problem, the demands of the latent heat of vaporization can be evaluated and located in time and space.

## 6  Discussion and conclusions

The evaporation front has been defined as an interface separating two different zones, wet and dry, inside or upon the boundary of a porous medium domain. The exact definition of these zones, presented in this paper, is based on the form of water they contain. Subsequently, the law of the evaporation front motion was formulated in the form of the vector equation (5). Since the law is based on the complete mass balance of water, i.e. liquid water and water vapor, it holds generally and does not need any additional account of energy. The laws of heat transfer and heat balance do not affect the presented equations which define the





evaporation front motion. On the contrary, solving problems that are fully determined by water transport data, the equations of the evaporation front motion can give certain insight into the energy requirements of such processes, e.g. the final part of section 5.

Smits et al. (2011) and Nuske et al. (2014) studied the process of evaporation from soils with the particular attention the phase change, and found that nonequilibrium models yield better agreement with experimental data than equilibrium models. The nature of the phase change process does not affect the results presented here directly, since the equation of the evaporation front motion requires other kind of data. The process of phase change enters Eq. (5) through its actual effect on the transport of water. On the other hand, the constitution laws like Darcy's law or the retention curve, that may be utilized when solving problems with Eq. (5), are equilibrium laws. In the example presented in section 5, equilibrium laws were utilized. However, the governing equation (5), being general, makes it possible to use nonequilibrium laws as well. Mls (1999) presented a general nonequilibrium approach to two-phase systems that keeps Darcy's law valid.

Lehmann et al. (2008) and Or et al. (2013) investigated the process of evaporation from the top of an initially saturated vertical column. They introduced the term characteristic length as the distance between the surface and the receding drying front (interface between the saturated zone and the unsaturated zone) and described different stages of the evaporation process. No evaporation front was introduced. In virtue of the present theory, the evaporation front cannot move in the positive direction of $\boldsymbol{\nu}(t,\boldsymbol{\xi})$ if

$$\boldsymbol{\xi}(t) \in \overline{\Omega_t^w} \cap B_t^d,$$

i.e. if $\boldsymbol{\xi}(t)$ is a point of a "dry from outside" part of the domain boundary. Hence, see Eq. (4), the front does not move until the condition

$$\Big(\big(\rho\,\boldsymbol{w}(t,\boldsymbol{\xi}) + c^w(t,\boldsymbol{\xi})\,\boldsymbol{v}^w(t,\boldsymbol{\xi}) + \boldsymbol{b}^w(t,\boldsymbol{\xi}) - \boldsymbol{b}^d(t,\boldsymbol{\xi}) - c^d(t,\boldsymbol{\xi})\,\boldsymbol{v}^d(t,\boldsymbol{\xi})\big), \boldsymbol{\nu}(t,\boldsymbol{\xi})\Big) < 0$$

is satisfied. In the simplified one dimensional case, Eq. (11), this condition reads

$$\big(\rho\,w(t,\xi) + b^w(t,\xi) - b^d(t,\xi)\big)\,\nu(t,\xi) < 0, \tag{21}$$

where $\nu(t,\xi) = 1$ if $x \in \Omega_t^w \Rightarrow x < \xi(t)$, and vice versa. Under conditions of sufficiently small values of $|b^d|$, condition (21) is not satisfied for a period, and the evaporation front does not move. Consequently, the evaporation rate does not change significantly. In the solved problem, the chosen initial conditions and the size of $\varepsilon$ make (21) valid even at $t = 0$, and the front moves from the very beginning of the process. If the flux density of liquid water from inside of the wet zone exceeds the flux density $b(t,L)$, either $\theta(t,L)$ increases or, being $\theta(t,L) = \theta_{sat}$, flux density $\rho\,w(t,L) - b(t,L)$ of liquid water discharges out of the porous medium domain.

The characteristic surface layer $\varepsilon$ was found experimentally, see (Slavík et al., 2020), and accepted in this paper as a part of the measured data. Note that Song et al. (2018) studied similar problems and introduced also a special diffusion layer outside the porous medium. From the viewpoint of moving front equations, the characteristic surface layer prevents infinite value of function $b^d$ at $x = 0$ and $t = 0$ which may be obtained when solving a problem with equation (8). This possibility origins in



the fact that the equation is a balance equation that contains an equilibrium law – the Fick law; for more on this problem and an alternate approach see (Mls and Herrmann, 2011).

The process of evaporation alone does not determine the direction of the evaporation front motion. Since the denominator
of the right-hand side of (5) is positive, the direction of the evaporation front motion is determined by the sign of the scalar product in the numerator. Consequently, both the processes of wetting or drying (increasing or decreasing the wet zone) can take place while evaporating water out of the profile; compare also Eqs. (13) and (20).

The presented problem example and its numerical solutions were aimed at showing the ability of the theory to simulate real processes, not at getting an optimized agreement. Most of the measured parameters of the solved problem were obtained
independently of the experiment. Only the pressure head and water flow data shown in Figures 2 and 3 were measured during the experiment and utilized as the imposed boundary conditions of the problem. The concentration of water in the gaseous phase, the functions $c_f$ and $c_e$, and the initial values of functions $h$ and $w$ were not measured but chosen. For the sake of their simple interpretation by means of acceptable values of temperatures and relative humidities, $c_f$ and $c_e$ were kept constant. No method of fitting was applied and a different choice of functions $h_i, w_i$ and constants $c_e$ and $c_f$ can give a better agreement
between the measured and computed values.

The presented theory is now prepared to prove its reliability on such problems that are fully documented and to be used when solving a wide range of problems of evaporation from a rock or soil profile.

*Author contributions.*  Jiří Mls is the author of this paper

*Competing interests.*  The author declares that he has no conflict of interest.

*Acknowledgements.*  This study was cofinanced by the Czech Science Foundation under Grant No. 19-14082S.



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



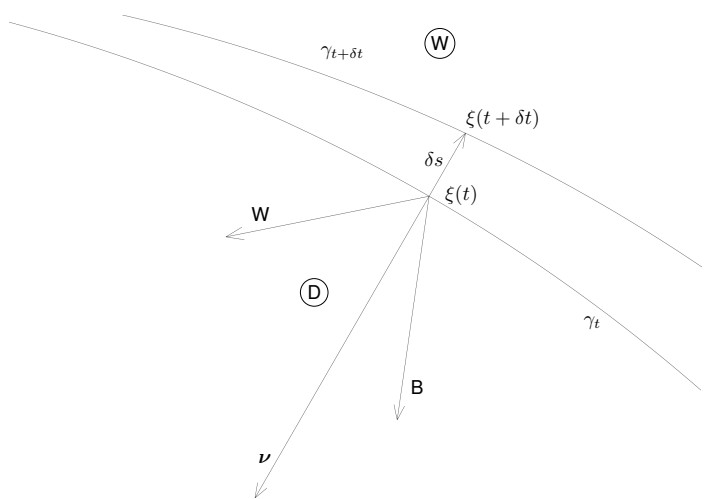

**Figure 1.** Evaporation front motion; $W$ is the liquid water flux density and vapor advection in the gaseous phase, $B$ is the water vapor flux density by diffusion in the gaseous phase, $\gamma_t$ is the evaporation front position at time $t$, $\nu$ is the unit normal to the front pointing out of the wet zone, $\xi(t)$ is the position of a chosen point upon the front at time $t$.





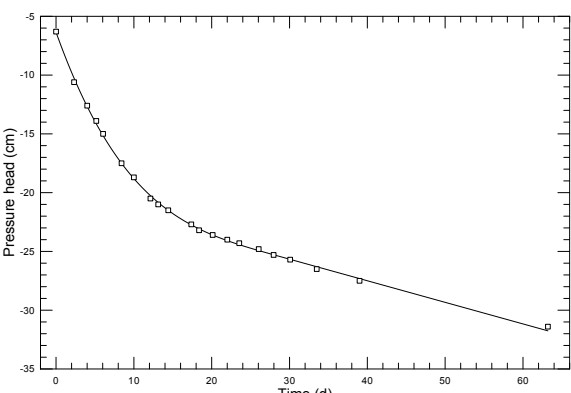

**Figure 2.** The $x = 0$ boundary condition – the pressure head at the boundary. The squares show the measured values, the smooth function is their approximation $h_0(t)$ that was used in the solved example.



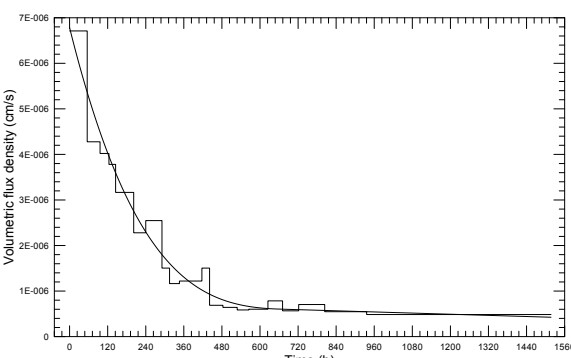

**Figure 3.** The $x = 0$ boundary condition – the volumetric flux density at the boundary. The step function shows the measured values, the smooth function is its approximation used in the solved example.





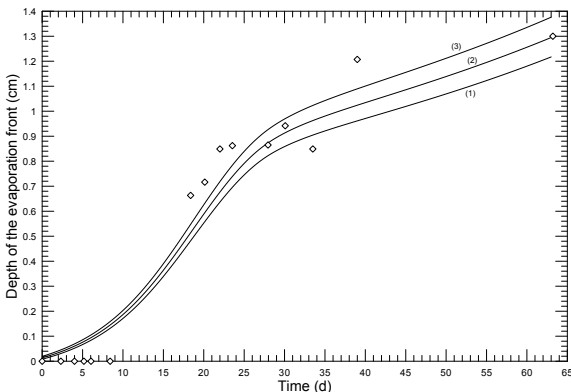

**Figure 4.** Evaporation front motion. $\diamond$: measured positions, solid lines: numerical simulations, (1) $c_f = 2.27 \times 10^{-5}$, $c_e = 9.35 \times 10^{-6}$, (2) $c_f = 2.41 \times 10^{-5}$, $c_e = 9.94 \times 10^{-6}$, (3) $c_f = 2.56 \times 10^{-5}$, $c_e = 1.06 \times 10^{-5}$, all in g/cm$^3$.