# Peer review of "EVAPORATION FRONT AND ITS MOTION"

_Hydrology and Earth System Sciences, 2021_

## Referee Comment (RC3)

The article is devoted to the definition of the evaporation front by means of porous media characteristics and the formulation the law of its motion in more or less general case of a water transport. In my opinion this topic is relevant and important for research. The evaporation front is considered as a sharp surface. First, the author generally defines the evaporation front with the help of intersection of dry and wet zones of the water-vapor-gas flow in question and their outer boundaries. Then, using the law of conservation of mass, the law of motion of the evaporation front is presented in a fairly general case. The unknown fluxes are determined from the basic equations, which express the fundamental laws in the field of a porous medium. The rest of the paper is devoted to the formulation of one dimensional problem and consideration of some special cases. However, the paper itself is written somewhat carelessly. I was not able to understand the derivation of the formula (5), which gives the main result of the paper. What do the designations $n$ and $\theta$ mean? What is in parenthesis? Scalar product with normal? In any case, the author should carefully clarify equation (3) and define the notations involved. In this context, I am finding the possibility of accepting the paper for publication as at least marginal. The paper needs the major revision, and then a new review for the possibility of publication.

---

## Author Comment (AC1)

However, the paper itself is written somewhat carelessly. I was not able to understand the derivation of the formula (5), which gives the main result of the paper.

I suggest to make the following change (blue color) of the text and to rearrange members of Eq. (3) so that the correspondence between them and the newly inserted text is clearly visible.

Let $\boldsymbol{\xi} = \boldsymbol{\xi}(t)$ be a point upon the evaporation front at time $t$; $\boldsymbol{\xi} \in \gamma_t$. Let $\delta t$ be an elementary time step and $\delta S$ an elementary surface surrounding the point $\boldsymbol{\xi}$, $\delta S \subset \gamma_t$. Denote by $\boldsymbol{\nu}(t, \boldsymbol{\xi})$ the unit normal vector to $\gamma_t$ at point $\boldsymbol{\xi}$, oriented out of the wet zone, and by $\delta s$ the distance between the positions $\gamma_t$ and $\gamma_{t+\delta t}$ at $\boldsymbol{\xi}$. Then the flow and transport of water coming to the elementary surface $\delta S$ from the wet zone, $\rho \boldsymbol{w} + \boldsymbol{b}^w + c^w \boldsymbol{v}^v$, pushes the front towards the dry zone, and the transport of water out of $\delta S$ into the dry zone, $\boldsymbol{b}^d + c^d \boldsymbol{v}^d$, pushes the front towards the wet zone. The excess of water coming to $\delta S$ during the time interval $\delta t$, $(\rho \boldsymbol{w} + \boldsymbol{b}^w + c^w \boldsymbol{v}^w - \boldsymbol{b}^d - c^d \boldsymbol{v}^d, \boldsymbol{\nu})$, is compensated by the mass of water $\delta s\, \delta S(\rho\, \theta + c^w(n - \theta) - n\, c^d)$ shifting the surface $\delta S$ to its new position distant by $\delta s$ in the direction $\boldsymbol{\nu}$. Here and in the sequel, $(\boldsymbol{u}, \boldsymbol{v})$, where $\boldsymbol{u}, \boldsymbol{v}$ are two vectors, denotes the scalar product. This account gives the balance equation

$$\delta S\, \delta t\big(\rho\, \boldsymbol{w}(t, \boldsymbol{\xi}) + \boldsymbol{b}^w(t, \boldsymbol{\xi}) + c^w(t, \boldsymbol{\xi})\, \boldsymbol{v}^w(t, \boldsymbol{\xi}) - \boldsymbol{b}^d(t, \boldsymbol{\xi}) - c^d(t, \boldsymbol{\xi})\, \boldsymbol{v}^d(t, \boldsymbol{\xi}), \boldsymbol{\nu}(t, \boldsymbol{\xi})\big)$$
$$= \delta S\, \delta s\big(\theta(t, \boldsymbol{\xi})\, \rho + (n - \theta(t, \boldsymbol{\xi}))c^w(t, \boldsymbol{\xi}) - n\, c^d(t, \boldsymbol{\xi})\big) \tag{3}$$

It is not necessary to make any changes in s Eqs. (4), (5) and (6), I only suggest to remove the superfluous brackets.

I believe that Eq. (5) does not require any further comment, since it was derived from Eq. (3) which is now explained in the blue colored lines.

**What do the designations n and _theta mean?**

I suggest not to use the symbol _theta (water content) in line 87, and to introduce and define it in line 92.

I suggest to introduce and define the symbol $n$ (porosity) in the sentence "Denote by ...", line 120.

**What is in parenthesis? Scalar product with normal?**

Yes, it is the scalar product. The notation is now explained in suggested blue colored text above.

---

## Author Comment (AC2)

While, in general, the manuscript is well-written, during reading you often get the feeling that important information is missing between the different steps presented in the study. Maybe the author could provide some additional details, which may be too obvious for the author; however, they may not be that obvious for the interested reader.

I understand that this item concerns in particular the step of getting the balance equation, Eq.(3). In order to make the step clear, I suggest to insert the following lines (new text is colored blue) and to rearrange members of Eq. (3) so that the correspondence between them and the newly inserted text is clearly visible.

Denote by $\boldsymbol{\nu}(t,\boldsymbol{\xi})$ the unit normal vector to $\gamma_t$ at point $\boldsymbol{\xi}$, oriented out of the wet zone, and by $\delta s$ the distance between the positions $\gamma_t$ and $\gamma_{t+\delta t}$ at $\boldsymbol{\xi}$. Then the flow and transport of water coming to the elementary surface $\delta S$ from the wet zone, $\rho\,\boldsymbol{w} + \boldsymbol{b}^w + c^w\,\boldsymbol{v}^v$, pushes the front towards the dry zone, and the transport of water out of $\delta S$ into the dry zone, $\boldsymbol{b}^d + c^d\,\boldsymbol{v}^d$, pushes the front towards the wet zone. The excess of water coming to $\delta S$ during the time interval $\delta t$, $(\rho\,\boldsymbol{w} + \boldsymbol{b}^w + c^w\,\boldsymbol{v}^w - \boldsymbol{b}^d - c^d\,\boldsymbol{v}^d, \boldsymbol{\nu})$, is compensated by the mass of water $\delta s\,\delta S(\rho\,\theta + c^w(n-\theta) - n\,c^d)$ shifting the surface $\delta S$ to its new position distant by $\delta s$ in the direction $\boldsymbol{\nu}$. Here and in the sequel, $(\boldsymbol{u},\boldsymbol{v})$, where $\boldsymbol{u},\boldsymbol{v}$ are two vectors, denotes the scalar product. This account gives the balance equation

$$\delta S\,\delta t\big(\rho\,\boldsymbol{w}(t,\boldsymbol{\xi}) + \boldsymbol{b}^w(t,\boldsymbol{\xi}) + c^w(t,\boldsymbol{\xi})\,\boldsymbol{v}^w(t,\boldsymbol{\xi}) - \boldsymbol{b}^d(t,\boldsymbol{\xi}) - c^d(t,\boldsymbol{\xi})\,\boldsymbol{v}^d(t,\boldsymbol{\xi}), \boldsymbol{\nu}(t,\boldsymbol{\xi})\big)$$
$$= \delta S\,\delta s\big(\theta(t,\boldsymbol{\xi})\,\rho + (n - \theta(t,\boldsymbol{\xi}))c^w(t,\boldsymbol{\xi}) - n\,c^d(t,\boldsymbol{\xi})\big) \tag{3}$$

It is not clear to me why the author has selected to limit the testing of the developed theory to some unpublished experimental data that are "not sufficiently documented" (as stated in line 203), instead of using some other published experimental data that have available all the required information. In that case the author would have the opportunity to examine if the proposed theory has accounted correctly for all the phenomena that are involved or some additional mechanism may be possibly missing. For example, M. Prat (Int. J. Heat Mass Transf., 50, 1455-1468, 2007) has indicated that accounting for film flows results in predicting more accurate the liquid saturation within a porous medium during evaporation. Otherwise, the provided numerical example is simply a mathematical exercise.

Solving a problem with Eq.(5), the knowledge of its right-hand side is required at any time and position of the macroscale evaporation front. I have not found such a set of data that make it possible. Moreover and perhaps the most important reason for using the "not sufficiently documented" experiment is the measurement of the front motion using the color method. To be utilized as a substitute for in situ measurements, the theory needs a comparison with results obtained by a method applicable under field conditions. The example is provided in order to show that the theoretical problem of evaporation front motion is numerically solvable and that the comparison with the experimentally measured data is possible.

The possible effect of film flows on the definition of the evaporation front has not been discussed in the current study. The evaporation front in the presence of film flows has been discussed by Yiotis and coworkers, as well by Prat and coworkers (e.g. see the related references in the aforementioned ref of Prat).

The role of the film flow is of course important in problems of evaporation in porous media. The approach of the present paper takes the film flow into account through its effect upon the flow in the wet zone and neglects its presence out of it. This follows from the definition of the front in lines 73-75.

The quality if the figures needs to be improved (e.g., font-sizes in x-, y-axes needs to be increased)

I made new figures and increased the font size.

All symbols should be clearly defined at the point where they are first introduced in the manuscript (e.g., density porosity in Eq. 3).

I checked the text once more and suggest to add the missed definition of $\theta$, the water content, to the line

93 and not to use the symbol in line 87. I also suggest to introduce and define the symbol $n$ (porosity) in the sentence "Denote by ...", line 120.

All equations should be numbered consistently (about half of the equations are not numbered).

Though I think that it suffices to number only equations that are referred to, I am ready to give numbers to all equations if required.

In order for someone to be able to reproduce the simulations discussed in the current study, it is essential that all the parameters used in the study should be provided either in the text or preferably collected in a Table (Mualem and van Genuchten parameters, Diffusion coefficients, etc.)

I suggest to insert the values of the parameters between the sentences "Now the problem (13), (12) can be solved numerically." and "Figure (4) shows ...", line 291.

In lines 76-78 the author mentions that "we do not consider the temperature distribution and heat flow balance". Does the author considers completely isothermal conditions? This issue should be stated clearly. Does this particular assumption applies to all experimental conditions and different porous media? If not, which are the restrictions/limitations for using this assumption?

The sentence tries to express the fact that (without any assumptions concerning heat or temperature) sufficient knowledge of water flow and transport (i.e. both liquid phase water and gaseous phase water) defines the position of the evaporation front. Consequently, it even gives the rate of the latent heat of vaporization being spent upon the front, see the lines 306-308. Of course, it cannot give the temperature distribution.

In order to make it clear, I suggest to replace the paragraph "We do not consider ..." lines 76-78, by the paragraph We do not consider the temperature distribution and heat flow and balance, since, in virtue of its definition, the evaporation front results from the water transport data. Though unknown, the heat flow within the profile provides the latent heat of vaporization that is necessary for the evaporation resulting from the actual process of water transport.

In lines 293-295 the author presents 3 cases for c_f and c_e. Additional details should be provided on how these values were selected (temperature, pressure conditions etc.) and what is the source of these values.

The only given values were those presented in lines 271-272. To show that the chosen values satisfy the requirements, I suggest to insert When using the method by Mc Rae (1980) and choosing the relative humidity at the evaporation front and at the sample's surface as 100% and 40%, respectively, we get the following temperatures at the evaporation front and at the sample's surface: (1) 19.5°C and 20.0°C, (2) 20.5°C and 21.0°C, (3) 21.5°C and 22.0°C. between lines 295 and 296.

The doi reported for Sakai et al. (2011) corresponds to a Correction to a previous paper by those authors. The author might want to include the original paper as well in the reference list.

The doi was incorrect. I will change it to 10.1029/2010WR009866.